# Advances in D-Amino Acids in Neurological Research

**DOI:** 10.3390/ijms21197325

**Published:** 2020-10-03

**Authors:** James M. Seckler, Stephen J. Lewis

**Affiliations:** 1Department Biomedical Engineering, Case Western Reserve University, Cleveland, OH 44106, USA; 2Department Pediatrics, Case Western Reserve University, Cleveland, OH 44106, USA; sjl78@case.edu

**Keywords:** d-amino acids, brain, Alzheimer, schizophrenia, neurological disorders, N-methyl-D-aspartate (NMDA) receptor

## Abstract

D-amino acids have been known to exist in the human brain for nearly 40 years, and they continue to be a field of active study to today. This review article aims to give a concise overview of the recent advances in D-amino acid research as they relate to the brain and neurological disorders. This work has largely been focused on modulation of the N-methyl-D-aspartate (NMDA) receptor and its relationship to Alzheimer’s disease and Schizophrenia, but there has been a wealth of novel research which has elucidated a novel role for several D-amino acids in altering brain chemistry in a neuroprotective manner. D-amino acids which have no currently known activity in the brain but which have active derivatives will also be reviewed.

## 1. Introduction

D-amino acids were originally considered to be inert compounds that were not endogenous to living organisms [1]. Research in the mid-20th century revealed their presence in bacteria, plants, and higher organisms [1,2,3,4]. In the 1980’s and 1990’s, D-aspartate and D-serine were discovered in the human brain and quickly linked to Alzheimer’s Disease [5,6,7,8]. It was later determined that these amino acids play a role in Alzheimer’s disease by acting as agonists and co-agonists for the N-methyl-D-aspartate (NMDA) receptor respectively [9,10]. Since then, D-aspartate and D-serine have been areas of active interest in D-amino acid research, and slowly other D-amino acids have been discovered to be neuroactive, being absorbed from a variety of sources [10,11,12]. 

In this review, we have divided D-amino acids into three categories based on their currently known role in mammalian biology (Figure 1). The first is D-amino acids, which directly interact with the NMDA receptor [13,14,15,16,17]. These are the most studied amino acids, and they are focused on a number of well-established disease models such as Alzheimer’s disease and schizophrenia [15]. The second are D-amino acids, which have NMDA receptor independent activities in the brain [18,19,20,21,22]. In many cases, these effects have only been recently discovered and they work through no unified mechanism or disease model [19,21,22]. Finally, the third group consists of D-amino acids that are inert within the brain but have interesting derivative molecules or metabolites in the brain [23,24,25,26,27]. Table 1 and Table 2 list the D-amino acids discussed in this review and summarizes how they enter the body, their biologically active derivatives, and gives a short description of the amino acid.

### 1.1. Sources of D-Amino Acids

D-amino acids come from a variety of sources including endogenous racemases, microbial production, and ingestion. Racemases are enzymes that catalyze the conversion of an L-isomer to a D-isomer and they are key enzymes in the production of D-isomer amino acids in the body [82]. At this time, serine racemase is the only racemase known to exist in the human body and it plays a role in the production of D-aspartate and D-serine [34,83]. This enzyme plays a significant role in the physiology and pathophysiology of a variety of diseases discussed later in this review. Interestingly, while serine racemase plays a role in D-aspartate production, it does not exclusively control D-aspartate synthesis, strongly suggesting the presence of a yet to be discovered human aspartate racemase [34]. While D-aspartate and D-serine are produced endogenously in mammals, they are also absorbed from gut bacteria along with D-alanine, D-glutamate, and D-proline [84,85,86]. This is a growing body of research on how gut bacteria affect neurology and play a role in the development of various diseases such as Alzheimer’s disease and schizophrenia [87,88,89,90]. There is potential for a role of D-isomer amino acids produced by gut bacteria in the genesis and progression of these and other diseases. Finally, all D-isomers are present to some degree in food and medicine that we ingest [12,82,91]. This comes from a variety of places and was recently summarized in detail [82].

### 1.2. Elimination of D-Amino Acids

Neurologically active D-amino acids are tightly regulated within the brain by the D-amino acid oxidase (DAAO) and D-aspartate oxidase (DDO) [1,35]. These two oxidases catalyze the oxidative deamination of their respective substrates and allow them to be excreted by the kidneys [35,42]. Most importantly D-Serine and D-Alanine are metabolized by human DAAO, while D-aspartate and D-glutamate are metabolized by DDO [42,92]. As all four of these amino acids modulate the NMDA receptor, there has been significant interest in their regulation, which will be discussed in detail later in this review. The primary method for eliminating D-amino acids is that they are transported to the kidneys and metabolized before being excreted [93,94,95]. While both DAAO and DDO are expressed in the brain, the highest levels of expression for DAAO are in the liver and kidneys, and the highest level of expression for DDO is in the heart [28,96]. Interesting, there have been recent studies showing that amino acids such as D-alanine are simply excreted by the kidneys rather than being metabolized, calling into question the centrality of DAAO in D-amino acid regulation [29,30,95]. It is likely that all D-amino acids not metabolized by either DAAO or DDO and are simply excreted by the kidneys as their primary method of elimination in humans. As most of the active D-amino acids are thought to be regulated by a single enzyme, there has been a plethora of recent studies on DAAO activity and its effects on D-amino acids levels within the body [30,35,36,40,52,68,80,96]. These studies will be discussed in detail in the following sections.

## 2. NMDA Receptor Agonists or Co-Agonists

Four D-isomer amino acids play a particularly important role in the body as endogenous neurotransmitters or neuromodulators. These compounds primarily work through modulating the NMDA receptor either as an agonist at the glutamate (D-aspartate/D-glutamate) or the glycine binding site (D-alanine/D-serine) [97]. As the NMDA receptor requires both sites to be occupied before the NMDA receptor’s ion channel will open [98]. Interestingly, it has been shown the various co-agonists binding to the glycine binding site are required to activate specific NMDA receptors [17]. In particular, it has been shown that D-serine and not glycine, is the endogenous co-agonist the CA3-CA1 synapses [17]. This hints at a more complex story for the D-amino acids which interact with the NMDA receptor. These amino acids have been the subject of significant research. In particular, the study of these amino acids is important for understanding the pathogenesis of diseases which directly involve the NMDA receptor and indeed, most work has focused on their role in Alzheimer’s Disease and schizophrenia [99,100]. It is established that over-activation of the NMDA receptor triggers neurotoxicity and plays a significant role in the neurodegeneration of Alzheimer’s Disease [99]. Alternatively, there is a growing body of evidence that under-activation of the NMDA receptor plays a key role in the pathophysiology of schizophrenia [100]. While most work focuses on these two disease models, there has been a wealth of other research into these D-isomers, and it is summarized in the following section.

### 2.1. D-Alanine 

D-Alanine plays a complex role in brain chemistry [31]. It long been shown that D-alanine elevates in the brains of patients suffering from Alzheimer’s disease [13], and that D-alanine can be used to treat schizophrenia [32]. In both cases, this has been linked to its ability to modulate the NMDA receptor [31,32]. A recent study showed that D-alanine strongly correlated with the Alzheimer’s Disease Assessment Scale—Behavioral Subscale (hallucinations, delusions, agitation, sundown syndrome, etc.), but no correlation with the Cognitive subscale [33]. The relative effectiveness of D-alanine in treating schizophrenia is blunted by the fact that the body quickly metabolizes it after administration [29]. These results have triggered significant interest in the body’s ability to modulate D-alanine, and until recently it has been taken as fact that D-alanine is absorbed from bacteria and degraded by DAAO [28,29,30,33]. This has meant that there has been a great deal of interest in DAAO inhibitors to help bolster D-alanine levels in schizophrenic patients [28,29,30]. Indeed, this seems to be the case in mice [28], but experiments in dogs and monkeys have shown no significant difference between animals taking DAAO inhibitors and those taking nothing [29,30]. Further evidence suggests that D-alanine accumulates in the kidneys and is excreted rather than metabolized [95]. Hence, the question of how D-alanine is regulated in humans remains an open and important one.

### 2.2. D-Aspartate

D-Aspartate is an endogenous amino acid that is present in all animals, including humans [101]. It functions as a neurotransmitter in the brain and plays a marked role in brain development, learning and memory [16,37,101]. The synthesis method for D-aspartate in the human brain is unknown, this process has recently been discovered to involve serine racemase [34]. This was determined by measuring the levels of D-aspartate in the brains of serine racemase knockout mice. It was shown that these mice produce significantly less D-aspartate in their hippocampus and prefrontal cortex but levels remained unaffected in their cerebellum [34]. This strongly suggests that serine racemase plays a significant role in D-aspartate production, but that other, yet to be discovered enzymes, also are used to produce it. Once produced, D-aspartate can act on its own, be metabolized to NMDA, or degraded by DDO [102]. This pathway is still being actively studied, and indeed a recent study detailing mouse DDO has shown some significant differences between mouse and human DDO [35]. In particular, mouse DDO showed increased cross reactivity with D-proline, and decreased flavin adenine dinucleotide binding compared to its human counterpart [35].

In the body, D-aspartate serves to modulate the NMDA receptor and stimulate the release of L-glutamate [36,103]. In a recent study, the antipsychotic drug, Olanzapine, was able to elevate D-aspartate levels and stimulate L-glutamate release in the pre-frontal cortex by inhibiting DDO [36]. D-Aspartate is most strongly expressed during brain development and declines with age [38]. It has been recently shown that this process is regulated by increased expression of DDO, which prevents neurodegeneration brought on by over activation of the NMDA receptor [38]. D-Aspartate’s role as an NMDA receptor activator may play a role in the etiology of schizophrenia since it is established that schizophrenic disease models have low levels of D-aspartate in the brain [39]. A recent study determined the reason for these low levels is an over-expression of DDO in the prefrontal cortex [96]. This work measured levels of D-aspartate in post-mortem brain samples of patients suffering from schizophrenia and found abnormally low levels in the prefrontal cortex but not hippocampus of these patients [96]. Recent work into D-aspartate has clearly shown that it is regulated by DDO, but also hints at a deeper story and undiscovered enzymes, which both create and degrade D-aspartate in the cortex and hippocampus.

### 2.3. D-Glutamate

In recent years, D-glutamate has become a molecule of interest for its protective effects against the behavioral symptoms of Alzheimer’s Disease [33,40,41]. Specifically, Alzheimer’s Disease is marked by a decrease in circulating D-glutamate as well as decreased D-glutamate in the hippocampus. It has been suggested that the loss of D-glutamate leads to reduced NMAD receptor activation and hence a worsening of symptoms [14,33,40,41]. D-Glutamate is thought to be absorbed from food as well as produced in the gut by bacteria, and it has been suggested that gut bacteria derived D-glutamate could be a novel therapeutic to slow the progression of Alzheimer’s Disease [14]. It has been known for some time that D-glutamate is metabolized by DDO, but a recent study has shown that it is also converted to 5-oxo-D-proline in cardiac mitochondria by a novel D-glutamate cyclase [95].

### 2.4. D-Serine

D-Serine has long been known to be a co-agonist of the NMDA receptor, which can occupy the glycine binding site [11,15]. This strongly links D-serine to both Schizophrenia and Alzheimer’s Disease [99,100]. Free D-serine is primarily localized in the mammalian forebrain, along with the highest concentrations of NMDA receptor [104,105,106]. D-Serine is created by serine racemase by converting L-serine into D-serine, which it is metabolized by DAAO [42,43,83]. D-Serine has a beneficial role in the brain and recent studies have broadened this story by showing how D-serine levels are regulated to protect the brain from damage or become dysregulated, causing disease [107,108,109]. D-Serine aids in recovery from traumatic brain injury, and in fact it has been suggested that D-serine is used by the brain to heal from traumatic brain injury in the form of an up-regulation of D-serine release by astrocytes [44,110,111]. This upregulation of D-serine release is caused by a 3-phosphoglycerate dehydrogenase dependent serine shuttle and further work is required to fully elucidate the mechanism that triggers this process during brain injury [107]. D-Serine reverses compulsive alcohol intake in rats, suggesting a role in treatment of alcoholism [108,109]. This involves D-serine inhibition of a unique type of NMDA receptor, which are only active at hyperpolarized potentials [108,109]. These specialized NMDA receptors primarily exist in the prefrontal cortex and have been shown to be involved in compulsive alcohol consumption [112]. 

There is a growing body of evidence that D-serine protects against schizophrenia, and that its loss reduces activation of NMDA receptors, bringing about schizophrenic symptoms [45,46,113,114,115,116]. Indeed, a number of studies have found that giving schizophrenic patients high doses of D-serine mitigates negative symptoms of schizophrenia [45,117,118,119]. This is important because it has recently been shown that D-serine levels are significantly lower in humans suffering from Schizophrenia, probably caused by blunted expression of serine racemase and overexpression of DAAO [114]. D-Serine plays a role in the onset of Alzheimer’s Disease and dementias [33,43,47,120]. Indeed, overexpression of D-serine in the brain appears to be a contributing cause of Alzheimer’s Disease [33,43]. It has been suggested that D-serine is released by astrocytes due to inflammation and leading to neurotoxicity in early Alzheimer’s Disease patients [43,48,120]. It has also known that the increased levels of D-serine within astrocytes come from an overexpression of Serine Racemase, which suggests a novel potential drug target for fighting Alzheimer’s Disease [48]. Conversely, D-serine has recently been shown to be reduced in the cerebrospinal fluid and in the substantia nigra of Parkinson disease patients, pointing to a more complex role for D-serine and the NMDA receptor in neurodegenerative diseases [121]. The recent body of work on D-serine shows great potential for novel therapeutics, which regulated its levels in the brain.

## 3. Neuroactive D-Isomers without Direct NMDA Receptor Interaction

There is another class of D-isomers that are active in the brain but are neither endogenous nor particularly well studied. These D-isomers are absorbed by ingestion (e.g., food, medication, etc.) and act through a variety of disparate mechanisms [12]. The work which has been done to elucidate the actions of these D-isomers is summarized in the following section.

### 3.1. D-Isoleucine

Traditionally, D-isoleucine has been viewed as largely inactive in the brain and instead research has been focused on its role in regulating bacterial behavior [49]. It has been shown to regulate the alanine-serine-cysteine-1 (Asc1) transporter [18]. This transporter releases D-serine and glycine in neurons and D-isoleucine stimulates D-serine release while inhibiting glycine uptake in isolated cells [18]. Recently, Mesuret et al. [50] has shown that D-isoleucine reduces glycinergic currents in brainstem currents and promotes glycine release in the brainstem in an Asc1 dependent manner [50]. This work was further built upon to show that D-isoleucine can be used rescue the long-term potentiation deficit of aging rats, protecting them from age related memory decline [51]. This is of particular interest because they showed that Asc1 does not seem to significantly contribute to normal NMDA receptor-dependent cognitive decline and indeed helps to prevent it. This leads to many additional interesting hypotheses such as could D-isoleucine have a role in managing the effects of schizophrenia by inducing D-serine release without leading to long term cognitive decline.

### 3.2. D-Leucine

D-Leucine is an understudied amino-acid which until recently had had no known activity in eukaryotic cells [19,52,53,54]. In spite of this, it has long been known that D-leucine is tightly regulated in the brain by DAAO [52], and that it is present in food and produced by bacteria [52,53]. Recently, it has been found that D-leucine is a potent anti-seizure drug that works against induced seizures but fails to prevent chronic epileptic seizures [19,122]. This seems to be because D-leucine is metabolized too quickly in the brain [52]. The same group attempted to identify the receptor that D-leucine acted through and identified the D-amino acid activated taste receptor type 1 member 2/3 (TAS1R2/R3) [122]. This was a good candidate due to previous work showing that other ligands of this receptor have anti-seizure properties [122]. However, TAS1R2/R3 knockout mice were found to be partially protected from induced seizures [19], suggesting a more complex role for this receptor and that the anti-seizure effects of D-leucine act through a yet undiscovered receptor in the mammalian brain.

### 3.3. D-Phenylalanine

D-Phenylalanine has been used in research for its ability to activate carbonic anhydrases [20]. Recent work using D-phenylalanine involved using it to probe the role of carbonic anhydrases in memory formation [20,55]. This work found that carbonic anhydrases in the hippocampus works to potentiate object recognition and fear extinction memory formation [55,123]. This was shown by the ability of D-phenylalanine to enhance memory formation when administered and the ability to carbonic anhydrase inhibitors, which could cross the blood brain barrier to abolish this effect [55,123]. This suggests that D-phenylalanine and carbonic anhydrase could play a broader role in memory formation.

### 3.4. D-Threonine

There has been little recent research on D-threonine in the brain. The sole study has shown that D-threonine is transported into rat hippocampal slices, where it causes a potentiation of currents within those slices [21]. This occurs by a yet to be determined mechanism which mimics taurine potentiation but is independent of the NMAD receptor [21,124,125]. Interestingly, it seems that many amino acids cause a similar potentiation including: L-alanine, D-alanine, L-glutamine, glycine, L-histidine, L-serine, D-serine, taurine, and L-threonine [21]. It is known that taurine’s potentiation acts partially through the mitochondria, and further work would be needed to ascertain if D-threonine acts through improving mitochondrial function [125].

### 3.5. D-Tyrosine

D-Tyrosine has no established activity in the brain but has recently been shown to be effective as a tyrosinase inhibitor, preventing melanin formation [22,56]. This is particularly interesting as tyrosinase has been implicated in neurodegenerative diseases such as Parkinson’s Disease [126,127]. It has also been suggested that tyrosinase inhibitors can have neuroprotective effects by preventing the overproduction of dopamine [128]. This suggests that D-tyrosine could have a neuroprotective effect in degenerative disease models such as Parkinson’s Disease, and it may warrant further research into this molecule in animal models.

## 4. Inactive D-Amino Acids with Interesting Derivatives or Other Activities

The final group of D-isomers are those which are thought to be inert in the brain by themselves, but when chemically modified in some way produce a compound with pronounced therapeutic potential. This involves a variety of neuroprotective compounds, which protect against brain injury, opioid overdose and cell death. These compounds are all novel and act through a variety of relatively poorly understood mechanisms, which are actively being studied.

### 4.1. D-Arginine

D-Arginine has and continues to be used as a negative control for inducible nitric oxide synthase (iNOS) activation [57] There has been a great deal of recent research on the neuroprotective effects of poly-D-arginine, particularly 18 amino acid poly-D-arginine-peptides (R18D) [58,59,60,61,62,129,130,131,132,133,134,135]. These R18D peptides use D-arginine to make them protease resistant and have been shown to be neuroprotective in experimental brain ischemic and/or hypoxic injury models [58,59,60,61,62,129,130,131,132,133,134,135]. This is thought to work through R18 uptake into cells and significantly shifting the electric potential of the cell causing a reduction in cell surface glutamate receptor levels and excitotoxic Ca2+ influx [23,61,63,130,136,137], and by stabilizing mitochondria, and reducing oxidative stress [132,133,134,135]. Other groups have also found that poly-arginine peptides are about to inhibit neuronal cell death by reducing stress-induced hyperpolarization of the mitochondria [63,138,139,140,141]. Additionally, poly-D-arginine peptides have been shown to be of use in gene therapy as non-viral carriers of genes [142,143,144,145]. This is due to their promotion of cell survival, allowing for introduction of a new gene while simultaneously preventing cellular stress [142,143,144,145]. This shows that D-arginine’s neuroprotective effects make it a useful tool for decreasing infarct volume after stroke, ischemic or hypoxic damage to the brain.

### 4.2. D-Cysteine

D-Cysteine is inert by itself but is still established to have neuroprotective properties within the brain [24,64,146]. It has been previously thought that these properties stem from the ability of D-cysteine to be metabolized to hydrogen sulfide (H2S) via conversion to 3-mercaptopyruvate by DAAO and then further conversion to H2S by 3-mercaptopyruvate sulfurtransferase [24,64,65,146]. This pathway is thought to be the reason that while D-cysteine is readily absorbed from food, it is seldom detected in the brain or blood stream [65]. There has been recent work to show that D-cysteine promotes the dendritic development of cerebellar Purkinje cells by a H2S mechanism [147]. Furthermore, it has been recently shown that human DAAO shows the greatest affinity for D-cysteine in spite of D-serine being expressed in the brain at much higher levels [42]. There has also been some dissenting work in this field, which hints at a more complicated story. D-Cysteine protects astrocytes from proteotoxicity through a mechanism identical to N-acetyl-L-cysteine, which cannot be degraded to H2S [66]. Furthermore, it has been shown that D-cystine dimethyl ester acts as a respiratory stimulant to reverse opioid induced respiratory depression again through a H2S independent mechanism [67]. Taken together this strongly suggests that D-cysteine, or closely related molecules, can act in a beneficial manner through a variety of direct and indirect biochemical processes.

### 4.3. D-Histidine

There has been no recent research on D-histidine in the brain or sensory systems. This is probably due to the fact that D-histidine is thought to be poorly transported into the brain and is inert upon entry [148,149,150]. It could be an interesting line of inquiry if there are any disease models which upregulate D-histidine into the brain or related tissues, then D-histidine could be used as an inert histochemical marker. 

### 4.4. D-Lysine

D-Lysine has long been known to be mostly inert in the brain, while being metabolized by DAAO [68,151,152]. In recent years, a modified version of D-lysine, oleoyl-D-lysine has been developed as an inhibitor of the glycine transporter-2 and the glycine receptor [25,69,70]. This makes oleoyl-D-lysine a potent non-opioid analgesic which works by blocking glycine transport and shows a great deal of potential for the treatment of chronic pain [25,153]. It would be interesting to see how oleoyl-D-lysine or a modified form of it could be applied to schizophrenia research, as oleoyl-D-lysine also inhibits glycine transporter-1, a protein which has a therapeutic role in the treatment of schizophrenia [25,154].

### 4.5. D-Methionine

D-Methionine has no known activity in the brain but protect against toxicity of a variety of drugs as well as protecting against noise-induced hearing loss [71,155,156]. A recent study found that D-methionine is transported into the brain using system L and the Asc transporter [72]. In this study, D-methionine was able to concentrate in the brain to significantly higher levels than L-methionine suggesting that there is no endogenous system for metabolizing D-methionine in the brain [72]. Recently, D-methionine was found to protect against ototoxicity in a variety of drug models by preventing the loss of connexin 26 and connexin 30 making it an interesting molecule for preventing drug induced cell death during pharmaceutical treatment [73,74,157].

### 4.6. D-Proline

D-Proline is largely inert in the brain in spite of readily passing through the blood brain barrier, and it has been used as a negative control in recent studies of L-proline [75,158]. This is because it has long been known that D-proline is rapidly converted to L-proline in the brain before being incorporated into proteins [158]. There has been interest in using a modified form of D-proline, Cis-4-[18F]fluoro-D-proline (D-cis-FPro), to detect and track the progression of cancer and neuro-degenerative conditions within the brain [26,76,77,78,159,160]. This technique uses positronic emissions tomography (PET) scanning methods to imagine the location of D-cis-FPro within the brains of living animals and humans [159]. D-Cis-FPro has been shown to be a marker for inflammation-associated degeneration that occurs in parkinsonian syndromes. D-cis-FPro has also been used to detect cell dead in human brain tumors after multimodal treatment [26]. This employed PET to imagine D-cis-FPro inside of the brains of living human beings before and after treatment for a variety of brain tumors [26]. Finally, there has been interesting work employing D-proline mutations to the amyloid-β peptide for the treatment of Alzheimer’s disease [79]. Lin et al. [79] created a construct of the 10–40 amino acids of the amyloid-β peptide where every valine was mutated to a D-proline. This construct was shown to significantly reduce the accumulation of endogenously released amyloid-β in Alzheimer’s disease model mice [79]. 

### 4.7. D-Tryptophan

The role that D-tryptophan plays in the brain has been largely ignored, but in recent years there has been some work into its role as a metabolic substrate and the role of a related compound in reversing ketamine induced schizophrenia [27,80]. In one study [80], D-tryptophan was converted to L-tryptophan and kynurenic acid by DAAO in mouse brains [42,80]. Kynurenic acid has several neural activities including anti-excitotoxic and anticonvulsant, and acts as a non-competitive antagonist at the glycine binding site of NMDA receptors [161,162]. Kynurenic acid is also known to trigger interleukin-6 expression through the aryl hydrocarbon receptor, making it a pro-inflammatory signaling molecule [163]. This pathway has been implicated in a number of diseases including cancer and schizophrenia [163]. It is of no surprise then, that kynurenic acid levels are elevated in patients experiencing schizophrenia [164], and that 1-methyl-D-tryptophan reverses the physiological indicators of ketamine-induced schizophrenia by acting as an inhibitor to indoleamine 2,3-dioxygenase, the enzyme that generates kynurenic acid from both L- and D-tryptophan [27]. This research is part of a growing body of literature that treats schizophrenia as an autoimmune disease, whereby interleukin-6 is activated by the kynurenic acid pathway and significantly contributes to the symptoms of schizophrenia [27,165].

### 4.8. D-Valine

There has been little interest in D-valine over the years. It is known to inhibit the growth of fibroblasts in primary neuronal cell cultures [166,167]. In recent years, there has been some interest in N-[[1-(5-fluoropentyl)-1H-indazol-3-yl]carbonyl]-3-methyl-D-valine methyl ester (5F-ADB), a novel cannabinoid which uses D-valine methyl ester as a scaffold [81]. This drug can elicit severe psychotic symptoms in humans, sometimes causing death. It was found that 5F-ADB activated brain dopaminergic neurons through activation of cannabinoid 1-receptor [81]. Interestingly, the L-valine form of this drug while still toxic, appears to require a higher dosage before fatality occurs [168,169,170].

## 5. Conclusions

There has been a wealth of research into D-amino acids in the past five years with the vast majority of it focusing on the amino acids, which serve as agonists or co-agonists to the NMDA receptor (D-aspartate, D-serine, D-alanine, and D-glutamate). This work has focused primarily on the sources and metabolism of these D-amino acids and hints at a wealth of future work, which is still to come. In particular, the question of what novel enzymes will be discovered in the coming years which either serve to produce D-amino acids (i.e., human aspartate racemase) or serve to metabolize them as was hinted at in the recent studies on D-alanine and D-glutamate [28,29,95,96]. There is also much work to be done in elucidating the various roles of these D-amino acids in the onset of progression of Alzheimer’s Disease, as recent work has clearly shown that these compounds have distinct pathophysiological roles within the brains of Alzheimer’s Disease patients [33]. Additionally, it seems that the loss of D-serine and hence NMDA receptor activation can also lead to neurodegeneration in the form of Parkinson’s disease, hinting at a much more complex role for D-serine and other D-amino acids in neurodegenerative diseases as a whole [121,171]. We expect the bulk of research within the coming years to focus on how D-amino acids modulate the NMDA receptor and how this modulation can be used to treat schizophrenia, Alzheimer’s disease, and a variety of other pathologies.

Many D-amino acids previously thought to be inert in the brain have been recently shown to have activity. This includes: (a) the ability of D-isoleucine to stimulate D-serine release without adding to cognitive decline, (b) the discovery of D-leucine as an anti-seizure agent, (c) the ability of D-threonine to potentiate currents within the hippocampus, and (d) the ability of D-tyrosine to inhibit tyrosinase. This combined with the ability of D-phenylalanine to inhibit carbonic anhydrase adds to a growing body of literature for how D-amino acids can be neuroprotective in a wide variety of disease models.

The remaining D-amino acids discussed in this review have, at present, no established role in the brain whereas they are therapeutically useful as modifications of these D-amino acids are used as drugs. D-arginine, D-lysine, and D-proline all have derivatives, which are used in a beneficial manner precisely because their base compound is relatively inactive in the brain. Alternatively, D-tryptophan and D-cysteine are metabolized to other compounds in the brain, where their derivatives work as inhibitors of this metabolism (D-tryptophan) or in a completely novel way (D-cysteine). The other D-amino acids have little to no interesting activity in the brain and have little current research, but this may change in future.

On the whole, the recent advances of D-amino acids within the brain has revealed a wealth of neuroprotective and beneficial properties and warrants further study as in many cases, the mechanisms by which a particular amino acid works is either poorly understudy or outright unknown, and elucidating these mechanisms could bring about great strides in the field of neurophysiology in general and Alzheimer’s disease and schizophrenia research in particular.

## Figures and Tables

**Figure 1 ijms-21-07325-f001:**
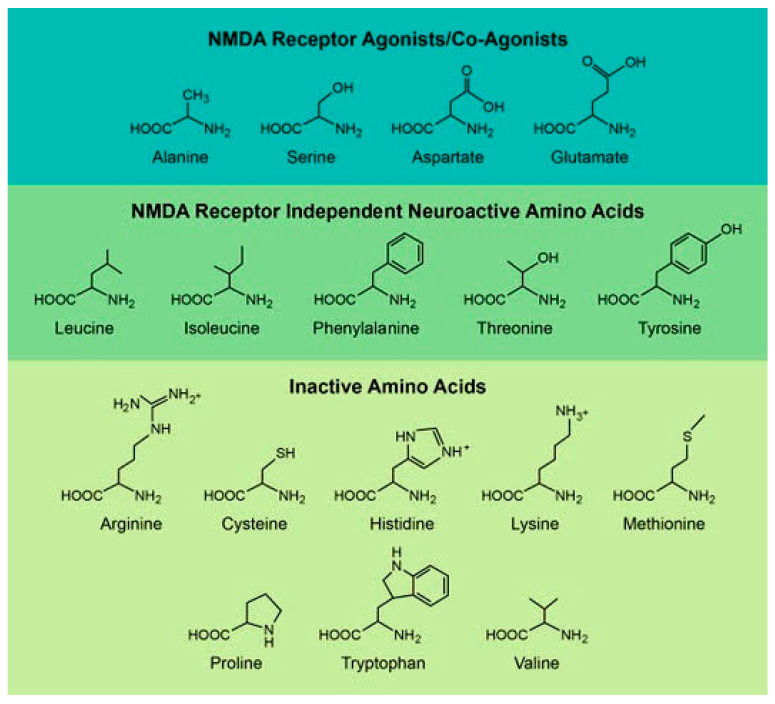
The chemical structures of the various D-amino acids which will be discussed in this review. We have divided these amino acids into three groups based on their in vivo activity.

**Table 1 ijms-21-07325-t001:** Summary of Neuroactive D-Amino Acids.

Amino Acid	Primary Source	Description	Reference
**NMDA Receptor Agonists or Co-Agonists**			
Alanine	Bacteria/Ingestion	NMDAR ^1^ co-agonist with links to Alzheimer’s behavioral changes	[13,28,29,30,31,32,33]
Aspartate	Bacteria/Racemase	NMDAR ^1^ agonist with various activities within the brain	[16,34,35,36,37,38,39]
Glutamate	Bacteria/Ingestion	NMDAR ^1^ agonist with links to Alzheimer’s behavioral changes	[14,33,40,41]
Serine	Bacteria/Racemase	NMDAR ^1^ co-agonist with links to Alzheimer’s Disease and Schizophrenia	[42,43,44,45,46,47,48]
**Active in the Brain through non-NMDA Receptor Pathways**			
Isoleucine	Ingestion	Stimulates D-serine release and inhibits glycine release in the brain	[18,49,50,51]
Leucine	Ingestion	Anti-seizure agent	[19,52,53,54]
Phenylalanine	Ingestion	Carbonic anhydrase activator	[20,55]
Threonine	Ingestion	Potentiates currents within the hippocampus	[21]
Tyrosine	Ingestion	Tyrosinase inhibitors	[22,56]

^1.^ NMDA receptor.

**Table 2 ijms-21-07325-t002:** Summary of Other Studied D-Amino Acids.

Amino Acid + B2	Primary Source	Derivatives	Description	Reference
Arginine	Ingestion	poly-18-D-arginine	Derivative is neuro-protective in brain injury models	[57,58,59,60,61,62,63]
Cysteine	Ingestion	D-cystine dimethylester	Metabolized to hydrogen sulfide in the brain, and the derivative can rescue opioid induced respiratory depression	[64,65,66,67]
Histidine	Ingestion	n.d. ^1^	No recently studied activity	None
Lysine	Ingestion	Oleoyl-D-lysine	Derivative is a non-opioid analgesic which acts via blocking glycine transport	[25,68,69,70]
Methionine	Ingestion	n.d. ^1^	No known activity in brain, but protects against ototoxicity in the ear	[71,72,73,74]
Proline	Bacteria/Ingestion	D-cis-Fpro	Derivative is a fluorescent probe used to monitor cancer progression	[75,76,77,78,79]
Tryptophan	Ingestion	1-methyl-D-tryptophan	Metabolized to kynurenic acid in the brain, and derivative can be used to inhibit kynurenic acid production	[27,80]
Valine	Ingestion	5F-ADB ^2^	Inhibits fibroblast growth, and the derivative is a synthetic cannabinoid with severe and often fatal side effects	[81]

^1.^none discussed, ^2.^N-[[1-(5-fluoropentyl)-1H-indazol-3-yl]carbonyl]-3-methyl-D-valine methyl ester.

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
