# Peer review of "Advances in D-Amino Acids in Neurological Research"

_ijms, 2020, doi:10.3390/ijms21197325_

Round 1

Reviewer 1 Report

Because of D-Amino acids play a role in Alzheimer’s disease by acting as agonists and co-agonists for the N-methyl-D-aspartate (NMDA) receptor respectively, the author aimed to give a concise overview of the recent advances in D-amino acid research as they relate to the brain and neurological disorders.

This manuscript interesting and includes a new information about the D-amino acids in neurological research. Nevertheless, this manuscript needs corrections and improvements before publishing is possible.

General points:

Please check and complete your list of abbreviations.

For better readability please add some Figures to your manuscript: for example, the structure of the different D-Amino Acids or mechanism of action of the different D-Amino Acids  and of course, everything what you want.

Special points:

Keywords: Please add also to keywords: neurological disorders; N-methyl-D-aspartate (NMDA) receptor

Special points:

Introduction

The Introduction section should be improved by adding references (it is a review)

Lines 19-20: please add references at the end of this sentence.

Lines 24-26: please add references at the end of this sentence.

Lines 27-29:  please add references at the end of each these sentences.

Lines 29-34: please add references at the end of each these sentences.

Table 1. and Table 2.: please add for each amino acid as a last column the number of  correspondent references used in this manuscript according to list of references.   

Manuscript text

Lines 43-44: please add references at the end of this sentence.

Lines 59-61: please add references at the end of each these sentences.

Lines 66-68: please add references at the end of this sentence.

Lines 70-72: please add references at the end of this sentence.

Lines 77-78: please add references at the end of this sentence.

Line 80: please add references at the end of this sentence.

Line 90: please add references at the end of this sentence.

Lines 95-99: please add references at the end of each these sentences.

Line 122: please add references at the end of this sentence.

Lines 145-147: please add references at the end of this sentence.

Lines 149-151: please add references at the end of this sentence.

Lines 175-177: please add references at the end of each these sentences.

Lines 193-194: please add references at the end of this sentence.

Lines 197-198: please add references at the end of this sentence.

Lines 198-199: please add references at the end of this sentence.

Line 214: please add references at the end of this sentence.

Lines 231-235: please add references at the end of each these sentences.

Lines 237-239: please add references at the end of this sentence.

Lines 247-250: please add references at the end of each these sentences.

Lines 316-317: please add references at the end of this sentence.

Line 325: please add references at the end of this sentence.

Lines 342-345: you said: There is also much work to be done in elucidating the various roles of these D-amino acids in the onset of progression of Alzheimer’s Disease, as recent work has clearly shown that these compounds have distinct pathophysiological roles within the brains of Alzheimer’s Disease patients [42].

What about Parkinson’s disease or other neurodegenerative diseases? Please add to the Conclusion section.

Author Response

Because of D-Amino acids play a role in Alzheimer’s disease by acting as agonists and co-agonists for the N-methyl-D-aspartate (NMDA) receptor respectively, the author aimed to give a concise overview of the recent advances in D-amino acid research as they relate to the brain and neurological disorders.

This manuscript interesting and includes a new information about the D-amino acids in neurological research. Nevertheless, this manuscript needs corrections and improvements before publishing is possible.

General points:

Please check and complete your list of abbreviations.

We double checked our abbreviations and found no omissions

For better readability please add some Figures to your manuscript: for example, the structure of the different D-Amino Acids or mechanism of action of the different D-Amino Acids and of course, everything what you want.

We have added a figure (new Figure 1) that includes all of the chemical structures of interest.

Special points:

Keywords: Please add also to keywords: neurological disorders; N-methyl-D-aspartate (NMDA) receptor

We have added these keywords to the article

Special points:

Introduction

The Introduction section should be improved by adding references (it is a review)

Lines 19-20: please add references at the end of this sentence.

Lines 24-26: please add references at the end of this sentence.

Lines 27-29:  please add references at the end of each these sentences.

Lines 29-34: please add references at the end of each these sentences.

We have added references to the above sections

Table 1. and Table 2.: please add for each amino acid as a last column the number of correspondent references used in this manuscript according to list of references.  

We have added the suggested column 

Manuscript text

Lines 43-44: please add references at the end of this sentence.

Lines 59-61: please add references at the end of each these sentences.

Lines 66-68: please add references at the end of this sentence.

Lines 70-72: please add references at the end of this sentence.

Lines 77-78: please add references at the end of this sentence.

Line 80: please add references at the end of this sentence.

Line 90: please add references at the end of this sentence.

Lines 95-99: please add references at the end of each these sentences.

Line 122: please add references at the end of this sentence.

Lines 145-147: please add references at the end of this sentence.

Lines 149-151: please add references at the end of this sentence.

Lines 175-177: please add references at the end of each these sentences.

Lines 193-194: please add references at the end of this sentence.

Lines 197-198: please add references at the end of this sentence.

Lines 198-199: please add references at the end of this sentence.

Line 214: please add references at the end of this sentence.

Lines 231-235: please add references at the end of each these sentences.

Lines 237-239: please add references at the end of this sentence.

Lines 247-250: please add references at the end of each these sentences.

Lines 316-317: please add references at the end of this sentence.

Line 325: please add references at the end of this sentence.

We have added references to the above sections

Lines 342-345: you said: There is also much work to be done in elucidating the various roles of these D-amino acids in the onset of progression of Alzheimer’s Disease, as recent work has clearly shown that these compounds have distinct pathophysiological roles within the brains of Alzheimer’s Disease patients [42].

What about Parkinson’s disease or other neurodegenerative diseases? Please add to the Conclusion section.

We have added this line to the text to expound out discussion to Parkinson’s disease and other NMDA receptor mediated neurodegenerative diseases.

“Additionally, it seems that the loss of D-serine and hence NMDA receptor activation can also lead to neurodegeneration in the form of Parkinson’s disease, hinting at a much more complex role for D-serine and other D-amino acids in neurodegenerative diseases as a whole [93,176].”

Reviewer 2 Report

This article is a comprehensive review of research on the role of D-amino acids in neurological function and dysfunction. The author presents an interesting classification of D-amino acids focused on whether they interact or not with NMDA receptors. Those that interact with NMDA receptors are sub-classified in agonists or co-agonists depending on their site of interaction at NMDA receptors. The review also discusses the possible sources of D-amino acids. The review concludes by highlighting future areas of research, particularly in elucidating the role of D-amino acids in the onset and progression of Alzheimer’s disease and Schizophrenia. The review is exhaustive and interesting to read. There are some minor points that I would suggest:

1)            The classification of D-amino-acids acting on NMDA receptors as agonists or co-agonists is over-simplistic. It would be appropriate to mention that some of them, such as D-serine act as partial agonists or functional antagonist depending on the synaptic levels of glycine, which has a higher affinity for the glycine co-agonist site of the NMDA receptor

2)            Inline 158 the sentence indicating that NMDA receptors are “specialized receptors primarily exist in the prefrontal cortex” is not correct. NMDA receptors are throughout the brain, although with different subunit compositions.

3)            The author might include in his review the recent article by Nuzzo et al. The levels of the NMDA receptor co-agonist D-serine are reduced in the substantia nigra of MPTP-lesioned macaques and in the cerebrospinal fluid of Parkinson’s disease patients. Sci Rep 9, 8898 (2019). https://doi.org/10.1038/s41598-019-45419-1

4)            Inline 180 minor typo “Traditionally, D-isoleucine has been views” to viewed

Author Response

This article is a comprehensive review of research on the role of D-amino acids in neurological function and dysfunction. The author presents an interesting classification of D-amino acids focused on whether they interact or not with NMDA receptors. Those that interact with NMDA receptors are sub-classified in agonists or co-agonists depending on their site of interaction at NMDA receptors. The review also discusses the possible sources of D-amino acids. The review concludes by highlighting future areas of research, particularly in elucidating the role of D-amino acids in the onset and progression of Alzheimer’s disease and Schizophrenia. The review is exhaustive and interesting to read. There are some minor points that I would suggest:

1)            The classification of D-amino-acids acting on NMDA receptors as agonists or co-agonists is over-simplistic. It would be appropriate to mention that some of them, such as D-serine act as partial agonists or functional antagonist depending on the synaptic levels of glycine, which has a higher affinity for the glycine co-agonist site of the NMDA receptor

We tried to find the papers which the reviewer was referring to but could not (D-serine acting as a partial agonist or antagonist depending on synaptic levels of glycine). If the reviewer wishes to provide references, we would be happy to include a discussion of this work. We did broaden the discussion of D-serine some by the inclusion of the following text:

Interestingly, it has been shown the various co-agonists binding to the glycine binding site are required to activate specific NMDA receptors [17]. In particular, it has been shown that D-serine and not glycine, is the endogenous co-agonist the CA3-CA1 synapses [17]. This hints at a more complex story for the D-amino acids which interact with the NMDA receptor.”

2)            Inline 158 the sentence indicating that NMDA receptors are “specialized receptors primarily exist in the prefrontal cortex” is not correct. NMDA receptors are throughout the brain, although with different subunit compositions.

We have reviews the text to clarify that we meant only a specialized subset of NMDAR rather than NMDARs as a whole

This involves D-serine inhibition of a unique type of NMDA receptor, which are only active at hyperpolarized potentials [63,64]. These specialized NMDA receptors primarily exist in the prefrontal cortex and have been shown to be involved in compulsive alcohol consumption [65].”

3)            The author might include in his review the recent article by Nuzzo et al. The levels of the NMDA receptor co-agonist D-serine are reduced in the substantia nigra of MPTP-lesioned macaques and in the cerebrospinal fluid of Parkinson’s disease patients. Sci Rep 9, 8898 (2019). https://doi.org/10.1038/s41598-019-45419-1

We have added this review and a discussion of it in the text

Conversely, D-serine has recently been shown to be reduced in the cerebrospinal fluid and in the substantia nigra of Parkinson disease patients, pointing to a more complex role for D-serine and the NMDA receptor in neurodegenerative diseases [93].”

“Additionally, it seems that the loss of D-serine and hence NMDA receptor activation can also lead to neurodegeneration in the form of Parkinson’s disease, hinting at a much more complex role for D-serine and other D-amino acids in neurodegenerative diseases as a whole [93,176].”

4)            Inline 180 minor typo “Traditionally, D-isoleucine has been views” to viewed

We have adjusted the text

Round 2

Reviewer 2 Report

The author has correctly addressed the indication to limit the potential effect of D-Serine to certain NMDA receptor subtypes and brain areas

Author Response

Thank you very much for your kind suggestion.